# Antibiotic Knowledge, Antibiotic Resistance Knowledge, and Antibiotic Use: A Cross-Sectional Study among Community Members of Bangkok in Thailand

**DOI:** 10.3390/antibiotics12081312

**Published:** 2023-08-12

**Authors:** Atsadaporn Niyomyart, Susan Ka Yee Chow, Wunwisa Bualoy, Nipaporn Butsing, Xingjuan Tao, Xuejiao Zhu

**Affiliations:** 1Ramathibodi School of Nursing, Faculty of Medicine Ramathibodi Hospital, Mahidol University, Bangkok 10400, Thailand; wunwisa.bul@mahidol.ac.th (W.B.); nipaporn.but@mahidol.edu (N.B.); 2Faculty of Medicine, Macau University of Science and Technology, Macau SAR, China; susanky.chow@connect.polyu.hk; 3School of Nursing, Shanghai Jiao Tong University, Shanghai 200025, China; tao.xingjuan@shsmu.edu.cn; 4School of Nursing, Hangzhou Normal University, Hangzhou 311121, China; jj_ice@163.com

**Keywords:** antibiotic knowledge, antibiotic resistance knowledge, antibiotic use, community population in Bangkok, Thailand

## Abstract

This study aimed to explore antibiotic knowledge, antibiotic resistance knowledge, and antibiotic use among adults in Bangkok, Thailand. This is a secondary analysis of cross-sectional data generated from a sample of 161 individuals living in Bangkok. Participants completed an online self-administered questionnaire developed by the World Health Organization. Descriptive analysis, the chi-square test, and multiple logistic regression analyses were performed. The sample comprised more females (56.5%) than males (42.2%). The majority of responders (67.7%) were between the ages of 18 and 40. More than half of the respondents mistakenly believed that antibiotics could treat colds and flu (54.7% and 47.2%, respectively). About 54.7% were aware that antibiotic resistance could harm them and their families. The chi-square test results showed that the levels of education were associated with antibiotic knowledge (*p* = 0.012), antibiotic resistance knowledge (*p* < 0.001), and antibiotic use (*p* = 0.023). Multiple logistic regressions showed that respondents with at least a bachelor’s degree or higher had better knowledge of antibiotics. Respondents who worked in the profession had better knowledge of antibiotic resistance. Respondents with sufficient incomes were more likely to use antibiotics. Baseline data from the study will be useful in antibiotic stewardship and public health campaigns among Bangkok residents.

## 1. Introduction

Antibiotics are medicines to prevent and treat bacterial infections, reducing morbidity and mortality [1]. Despite the fact that antibiotics improve survival from bacterial diseases, antibiotic use is associated with an increased prevalence of antibiotic resistance globally [2,3]. Antibiotic resistance has become a global public health concern as the prevalence of antibiotic resistance leads to masking symptoms, treatment failure, drug resistance, higher medication cost, and adverse drug events including death [4]. In 2019, a systematic analysis of the global burden of bacterial antimicrobial resistance (AMR) among 204 countries and territories revealed that 4.95 million deaths were related to bacterial AMR, of which 1.27 million were caused by bacterial resistance [5]. By 2050, this number is estimated to rise to 10 million deaths per year if no action is initiated [6]. Antibiotic resistance has an impact not only on human health and lives but also on overall healthcare costs [7]. There is an estimate that antibiotic resistance will result in healthcare expenditures of 20 billion dollars per year in direct healthcare costs in the United States and up to 3.5 billion dollars annually in Europe, North America, and Australia [7,8].

In Thailand, at least 88,000 infected people are exposed to antibiotic resistance, and approximately 38,000 people die each year [9]. The direct costs associated with AMR in 2010 were about 70–170 million dollars, while indirect costs for morbidity and premature death totaled 1100 million dollars [9]. A health survey in 2014 among 19,468 Thai residents reported that half of the residents who experienced colds, diarrhea, and simple wounds used antibiotics for their illnesses [10]. According to the National Statistical Office of Thailand and the International Health Policy Program, eight percent of Thais have used antibiotics for fever or sore throat [10]. In fact, the residents can access most antibiotics from retail pharmacies without a prescription. The data showed that Thai residents spent 0.9% of their expenses on purchasing antibiotics in 2013, and it increased to 1.6% in 2014 [11]. Antibiotic resistance is a major problem in low- and middle-income countries (LMICs) due to the widespread use of antibiotics [12,13]. More than 50% of antibiotics are purchased without a prescription globally [14,15], and self-medication with antibiotics is becoming increasingly common. In Vietnam and Bangladesh, for instance, the percentage of antibiotics used without a prescription range from 38.0–55.2% and 45.7%, respectively [12,16]. Similarly, a licensed pharmacist can legally dispense antibiotics in Thailand, where residents can purchase most antibiotics from pharmacies without a prescription, potentially resulting in overuse or inappropriate use of antibiotics. In 2016, the Thai government endorsed its first National Strategic Plan 2017–2021, which was aligned with a Global Action Plan developed by WHO in 2015 to promote public awareness of antibiotic use and tackle antibiotic resistance [17]. Over the three-year period from 2017 to 2019, knowledge of AMR and antibiotic use increased by 0.6% from 23.7% to 24.3%, while human antibiotic consumption increased by 20.9% from 68.4% to 83.0% of the defined daily dose per 1000 inhabitants per day, in comparison to the goal of reducing antibiotic consumption by 20% and improving antibiotic knowledge and antibiotic resistance by 20% [18,19]. Indeed, inadequate knowledge of antibiotics is associated with irrational use of antibiotics within the community, resulting in antibiotic resistance [20,21,22,23].

This study aimed to explore antibiotic knowledge, antibiotic resistance knowledge, and antibiotic use among residents of Bangkok, which is the capital and most populated city of Thailand. To our knowledge, no study has been conducted on antibiotic use since the national strategic plan was announced, especially in the capital city where pharmacies are located everywhere. We expect the findings from this study can provide the future basis for antibiotics stewardship initiatives aligned with residents’ needs in Bangkok.

## 2. Results

### 2.1. Sociodemographic Characteristics

The majority of participants were predominantly female (56.5%), ranged in age between 31 and 40 years (34.2%), and never married (47.8%); 57.1% had a sub-degree or bachelor’s degree. More than half of the participants (57.8%) were individuals who lived in private housing (i.e., single-family homes, townhomes). The demographic characteristics are shown in Table 1.

### 2.2. Antibiotic Knowledge

Figure 1 presents antibiotic knowledge. Only 16.8% of respondents correctly answered that they should stop taking antibiotics after taking them as prescribed. However, 66.5% of respondents correctly answered that antibiotics given to a friend or family member should not be used for themselves even if they had the same illness. Despite having the same symptoms before, almost half (48.4%) of respondents were aware that they should not purchase or ask for the same antibiotics.

Figure 2 presents responses to conditions to be treated with antibiotics. Most respondents correctly identified bladder or urinary tract infections and skin or wound infections as illnesses that can be treated with antibiotics (70.2% and 67.7%, respectively); in contrast, only 57.8% of respondents correctly identified gonorrhea as a treatable condition with antibiotics. Moreover, large proportions of respondents mistakenly understand that conditions that are viral can be treated with antibiotics, notably colds and flu (54.7%) and sore throats (47.2%). In addition, more than a quarter of respondents answered “do not know” in response to measles (37.9%), HIV/AIDS (34.2%), and malaria (30.4%). Of the conditions tested, respondents recognized the correct treatment for headaches and body aches (50.3% and 56.5%, respectively).

### 2.3. Antibiotic Resistance Knowledge

Table 2 shows whether respondents were familiar with antibiotic resistance terminology. Most respondents have never heard of Superbugs (75.2%) and antimicrobial resistance (46.6%). In contrast, 85.1% of respondents are familiar with antibiotic resistance and 49.7% for antibiotic-resistant bacteria terms. In terms of antibiotic resistance knowledge, 54.7% of respondents correctly identify “Antibiotic resistance is an issue that could affect me or my family” as a true statement (Figure 3).

Figure 3 showed that the proportion of respondents who answered correctly that “Many infections are becoming increasingly resistant to treatment by antibiotics” is 54%. However, 70.8% of respondents think that the statement “Antibiotic resistance occurs when your body becomes resistant to antibiotics and they no longer work as well” is true, when this is, in fact, a false statement. In addition, a greater proportion (40.4%) of respondents also think that “Antibiotic resistance is only a problem for people who take antibiotics regularly” is true, whereas, in fact, it is a false statement. In relation to the statements which are most answered “do not know” are “Bacteria which are resistant to antibiotics can be spread from person to person” (39.1%) and “Antibiotic-resistant infections could make medical procedures like surgery, organ transplants, and cancer treatment much more dangerous” (42.2%).

### 2.4. Antibiotic Use

Bangkok residents have used antibiotics as follows: 37.9% in the last six months, 28% in the last 30 days, and 12.4% in the last year. Only 7.5% of respondents reported never using antibiotics, and 14.3% could not recall when they last used them (Figure 4).

### 2.5. Antibiotic Knowledge, Antibiotic Resistance Knowledge, and Antibiotic Use

Antibiotic knowledge, antibiotic resistance knowledge, and antibiotic use were significantly associated with respondents who had a higher level of education ($X^{2}=$ 8.897, *p* = 0.012; $X^{2}=26.739,$ *p* < 0.001; $X^{2}=$ 7.535, *p* = 0.023, respectively). Further, antibiotic use was found to be significantly related to income ($X^{2}=$ 5.965, *p* = 0.015). The knowledge about antibiotics as well as antibiotic resistance was associated with occupation ($X^{2}$ = 9.581, *p* = 0.008; $X^{2}$ = 17.073, *p* < 0.001, respectively). No significant difference was found for gender and age (Table 3).

### 2.6. Factors Associated with Antibiotic Knowledge, Antibiotic Resistance Knowledge, and Antibiotic Use

Multiple logistic regression analyses (Table 4) showed that education level was significantly associated with antibiotic resistance knowledge and antibiotic use, with respondents with bachelor’s degrees or higher having better knowledge and using antibiotics more often than those with high school diplomas or less. Occupation was significantly associated with antibiotic knowledge, with professional respondents having better antibiotic knowledge than those with other occupations. Income was significantly associated with antibiotic use, with respondents with adequate incomes being more likely to use antibiotics compared to those with inadequate incomes.

## 3. Discussion

This study provides insights into antibiotic knowledge, antibiotic resistance knowledge, and antibiotic usage among particular residents in Bangkok.

In terms of antibiotic knowledge, although a majority of the respondents correctly answered that antibiotics were used to treat bladder infections or UTIs (70.2%), more than half of the respondents (54.7%) believed that antibiotics can treat colds and flu. Further, nearly half (37.9%) did not know antibiotics could not treat measles. This may indicate that the respondents might not be well informed about antibiotics; they are therefore unable to distinguish between bacterial and viral infections. As found in prior studies, Thais (50.7%) believe antibiotics kill viruses, whereas Malaysians (77.0%), Indians (45.1%), Romanians (39.26%), and Italians (23.6%) believe antibiotics are effective in combating colds and flu [3,24,25,26,27]. Further, the present study showed that women with an education level higher than a bachelor’s degree, a professional job, and an adequate income were more knowledgeable about antibiotics than men. It should be noted that age and gender are not consistently regarded as having sufficient knowledge, and only education level and occupation were found significant. This was supported by a few recent previous studies that reported an association between educational level and knowledge of antibiotics [28,29,30]. Thus, targeted interventions tailored to the education level of specific groups should be considered to address knowledge gaps and misconceptions regarding antibiotics.

Regarding antibiotic resistance knowledge, a majority of respondents (70.0%) misunderstood that antibiotic resistance occurs when the body becomes resistant to antibiotics, and they no longer work. Furthermore, only 45.3% of respondents correctly identified antibiotic resistance as a danger to surgeries, organ transplants, and cancer treatments. In addition, the percentage of those who answered “I do not know” was almost the same as the percentage of those who answered correctly (42.2%). Another finding was that only 36% of respondents correctly stated that antibiotic-resistant bacteria can only be transmitted from person to person. These findings imply that there are some gaps as well as widespread misconceptions about antibiotic resistance. The results of our study are in line with the findings of the WHO survey, which found that 76% of respondents believed antibiotic resistance is caused by body resistance [31]. In addition to these findings, previous studies conducted in Cyprus and Singapore also reveal similar findings [31,32]. As with antibiotic resistance knowledge, women with more than a bachelor’s degree and professional work had better antibiotic resistance knowledge. The result is in line with the study carried out in Japan [33]. The findings showed that a lack of antibiotic resistance knowledge can result in misconceptions, which can lead to misuse and overuse of antibiotics, resulting in antibiotic resistance.

In regard to antibiotic use, more than three-quarters (78.3%) of the respondents reported having taken an antibiotic in their lifetime prior to our study. The result showed that the overall antibiotic usage was higher than the global antibiotic consumption (65%); however, the result is in line with LMICs such as Ukraine (80.0%), Egypt (80.0%), Kazakhstan (80.0%), Bosnia (79.0%), and Sudan (73.0%) [34,35,36]. This indicates that although Thailand has developed a national strategic plan to combat antibiotic resistance, this plan may not communicate effectively with regard to the use of antibiotics, specific risks of antibiotic overuse, and existing knowledge about antibiotic use behavior. Moreover, Bangkok has approximately 4676 pharmacies [37], where residents can easily purchase over-the-counter and cheap antibiotics (i.e., amoxicillin, norfloxacin, ciprofloxacin) without a prescription, so residents can access antibiotics and even stock up for future use without a doctor’s prescription. Nevertheless, it is important to note that our study recruited participants during the COVID-19 pandemic, so we do not know whether our participants have been exposed to COVID-19, which may make them more susceptible to infections and therefore use more antibiotics. Clearly, antibiotics overuse is a very challenging problem in Thailand, where there is a need for political commitment, regular structure, financial resources, data availability on antibiotic use, overuse, and misuse, and national drug policy to sustainably tackle antibiotic resistance [17]. Further, the government along with the healthcare sector must provide clear information on antibiotic use through online platforms as well as healthcare facilities; similarly, health professionals and pharmacy training are essential in ensuring that antibiotics are dispensed properly and raising awareness of the need for antibiotics.

Following the announcement of Thailand’s national strategic plan, this is the first survey to assess antibiotic knowledge, antibiotic resistance knowledge, and antibiotic usage among Bangkok residents. As a result, its findings provide baseline evidence as well as the need for ongoing political commitment and multisector engagement to address antibiotic resistance.

### Limitations

This study has some limitations. First, no causal inference could be drawn due to its cross-sectional study design. Second, self-reporting may lead to under-reporting socially desirable or over-reporting socially undesirable behaviors, resulting in bias. Third, our small sample size may not be representative of the residents in Bangkok, and a larger sample is needed in the future. Finally, this study only included participants who had access to a computer/mobile device and the Internet. Thus, these data may not be generalized to other demographic groups.

## 4. Materials and Methods

### 4.1. Study Design and Sample Sampling

This study was a secondary analysis of data obtained by a cross-sectional descriptive study entitled “how socioeconomic, health-seeking behaviors and educational factors are affecting the knowledge and use of antibiotics in four different cities in Asia” [38]. The primary aim was to investigate factors of antibiotic resistance knowledge. A more detailed description has been presented in the previous report [38]. For this study, we examined if sociodemographic characteristics predicted better or less antibiotic knowledge, antibiotic resistance knowledge, and antibiotic usage among Bangkok residents.

#### Inclusion and Exclusion Criteria

A sample of 161 participants was enrolled from different districts in Bangkok. Participants could participate in this study if they met the following inclusion criteria: (1) 18–65 years of age, (2) living in Bangkok, (3) having a job, and (4) being able to read, speak, and understand Thai. The survey was designed to gather information about knowledge of antibiotics and antibiotic resistance, and antibiotic use among Bangkok residents. The study protocol was approved by the hospital Institutional Review Board, in Thailand.

### 4.2. Sample Size

An estimation of the sample size for logistic regression analysis was based on the formula n = 100 + xi, where x is an integer and i is the number of independent variables [39]. An event per variable (EPV) of 10 was selected, and five sociodemographic variables were tested in the logistic model, so a minimum sample size of 150 was required. For this study, 161 participants would be an adequate sample size. The formula to calculate sample size is shown below.
n = 100 + xi
where n = sample size
x = integer (x = 10)
i = independent variable (i = 5)

### 4.3. Data Collection Method

Participants were invited to complete an internet-based survey between January and August 2021. The invitation explained the study, the principal investigator’s contact, the voluntary nature of the study, and the anonymous distribution of the survey. The participants who agreed to participate in this survey could access the electronic survey; those who did not agree to participate were directed out of the Google Form. All data were kept on a specifically designed data-collection form. Only the principal investigator (PI) could access the participant records.

#### 4.3.1. Questionnaire for Data Collection

The questionnaire consisted of two sections as follows: (1) sociodemographic characteristics (i.e., age, sex, income, level of education, income), and (2) Antibiotic Resistance: Multi-Country Public Awareness Survey—the use of antibiotics, antibiotic knowledge, and antibiotic resistance knowledge [20]. These questions were examined based on a three-point scale (yes, no, uncertain).

#### 4.3.2. Antibiotic Resistance: Multi-Country Public Awareness Survey

The original version of the Antibiotic Resistance Survey was designed in the English language by WHO. Permission to use the scales was obtained from the WHO and the Centre for Health Protection, Hong Kong. The back-translation was changed from English to the Thai language by a translator who is a Thai native speaker and knowledgeable of the English-speaking culture. A target language version was then translated back into the English version by another translator and sent it back to the developer for comprehensiveness, clarity, appropriateness, and/or cultural relevance of the research instrument [40]. Content validity was assessed by a panel of four experienced nurses who are knowledgeable about antibiotics. They rated the survey as complete and did not add or delete any item. For face validity, seven subjects indicated that the items of the survey were relevant, clear, easy to understand, and appropriate to use for the broader audience. A pilot study consisting of 53 participants completed surveys has been conducted to confirm the reliability of the scale. Cronbach’s alpha of antibiotic use was 0.62; none of the items could be deleted to improve the internal consistency of the instrument. Cronbach’s alpha was 0.83 for knowledge about antibiotics, and 0.83 for antibiotic resistance knowledge.

### 4.4. Data Analysis

Data were transferred to Microsoft Excel. The answers for antibiotic knowledge and antibiotic resistance knowledge items were coded as correct and incorrect answers. Each correct answer was assigned a score of “1” and other responses (incorrect or unsure responses) were given a score of “0”. Responses to the use of antibiotics were coded as “1” (in the last 30 days, six months, last year, more than a year ago) and other responses (never use or cannot remember) were coded as “0”. All data were coded, and missing data were checked and saved as a comma-separated values (CSV) file, which was then transferred to R version 3.6.3 for data analysis [41]. We used frequencies and percentages to describe sociodemographic variables and responses about antibiotic knowledge, antibiotic resistance knowledge, and antibiotic use. The median score based on responses to questions in the knowledge sections was used as the cut-off to dichotomize the continuous variable for use as the dependent variable in logistic regression analysis [42,43]. Respondents scoring higher than the median were defined as having “better knowledge of antibiotics and antibiotic resistance” [42,43]. Chi-square tests were performed to identify the correlation between them using the “chisq.test” function in R. Three multiple logistic regression models were performed using the “glm” function to determine whether sociodemographic factors are associated with antibiotic knowledge, antibiotic resistance knowledge, and antibiotic use. The significance level (α) was set at 0.05 for all statistical tests.

## 5. Conclusions

In the present study, we provide valuable baseline data on residents of Bangkok’s knowledge of antibiotics, antibiotic resistance, and antibiotic use, which will help establish public health strategies for raising awareness and promoting appropriate antibiotic use. A collaborative effort between clinicians, pharmacists, other healthcare professionals, drug companies, universities, and public health volunteers is necessary to monitor antibiotic consumption, antibiotic resistance, and prescription policies for antibiotics.

## Figures and Tables

**Figure 1 antibiotics-12-01312-f001:**
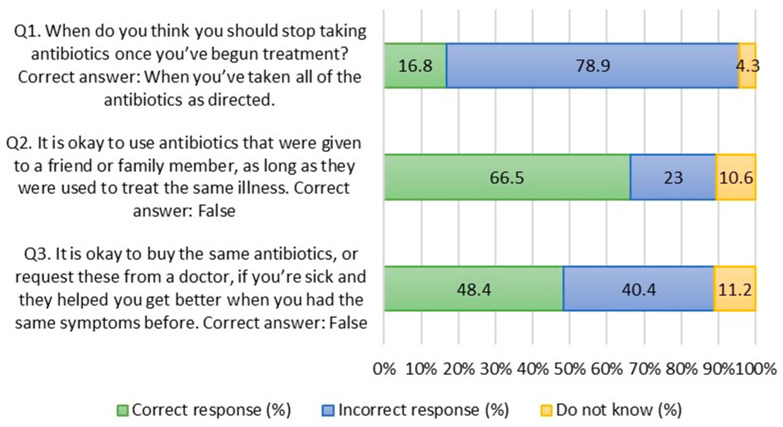
Antibiotic knowledge.

**Figure 2 antibiotics-12-01312-f002:**
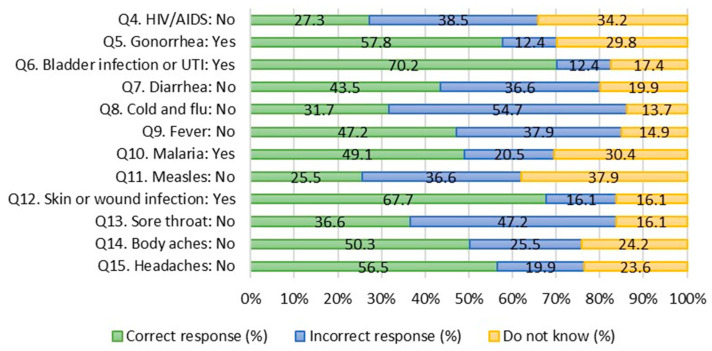
Conditions to be treated with antibiotics. Note. HIV = Human immunodeficiency virus; AIDS = Acquired immunodeficiency syndrome; UTI = Urinary tract infection.

**Figure 3 antibiotics-12-01312-f003:**
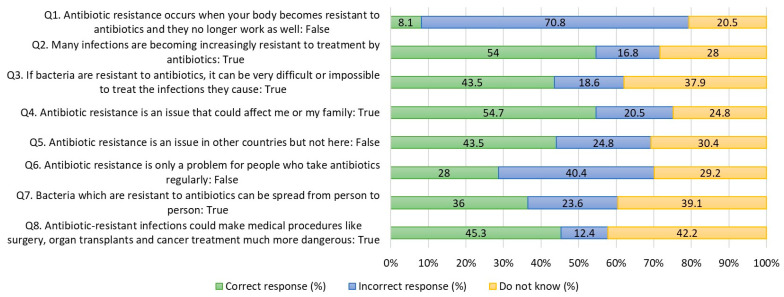
Antibiotic resistance knowledge.

**Figure 4 antibiotics-12-01312-f004:**
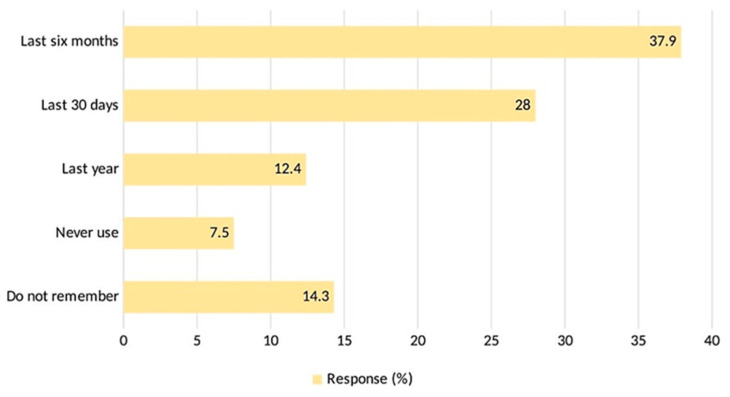
Antibiotic use.

**Table 1 antibiotics-12-01312-t001:** Sociodemographic characteristics of the residents in Bangkok (*N* = 161).

Variable	Participants *n* (%)
Age (year) ^a^	
18–30	54 (33.5)
31–40	55 (34.2)
41–50	37 (23.0)
51–60	12 (7.5)
61+	2 (1.2)
Gender	
Female	91 (56.5)
Male	68 (42.2)
Not identify	2 (1.2)
Marital status	
Never married	77 (47.8)
Married with child	71 (44.1)
Married and without a child	11 (6.8)
Divorced or Separated	2 (1.2)
Education	
Primary or below	3 (1.9)
Lower secondary (S1–S3)	10 (6.2)
Upper secondary (S4–S6)	39 (24.2)
Sub-degree or bachelor’s degree	92 (57.1)
Above bachelor’s degree	17 (10.6)
Income salary	
Very adequate	2 (1.2)
Adequate	78 (48.4)
Barely adequate	51 (31.7)
Not adequate	22 (13.7)
Very inadequate	8 (5.0)
Living accommodation	
Flat public/rental condominium	25 (15.5)
Flat/condominium owner	21 (13.0)
Single-family home/townhouse/shop house	93 (57.8)
Single room	8 (5.0)
Staff/student quarters	8 (5.0)
Other	6 (3.7)

Note. ^a^: *n* = 160.

**Table 2 antibiotics-12-01312-t002:** Antibiotic resistance terminology.

Questions	Yes	No	Do Not Know
	n (%)	n (%)	n (%)
Have you heard of antibiotic resistance?	137 (85.1)	16 (9.9)	8 (5.0)
Have you heard of Superbugs?	11 (6.8)	121 (75.2)	29 (18.0)
Have you heard of antimicrobial resistance?	61 (37.9)	75 (46.6)	25 (15.5)
Have you heard of antibiotic-resistant bacteria?	80 (49.7)	59 (36.6)	22 (13.7)

**Table 3 antibiotics-12-01312-t003:** Correlation between sociodemographic variables and antibiotic knowledge, antibiotic resistance knowledge, and the use of antibiotics.

Variables	ATB Knowledge			ATB Resistance Knowledge			ATB Use		
	Less n (%)	Better n (%)	$X^{2}$ (*p* = Value)	Less n (%)	Better n (%)	$X^{2}$ (*p* = Value)	No n (%)	Yes n (%)	$X^{2}$ (*p* = Value)
Gender			2.952 (*p* = 0.086)			0.833 (*p* = 0.361)			0.159 *(p* = 0.690)
Male	40 (58.8)	28 (41.2)		43 (63.2)	25 (36.8)		16 (23.5)	52 (76.5)	
Female	41 (45.1)	50 (54.9)		51 (56.0)	40 (44.0)		19 (20.9)	72 (79.1)	
Age			1.984 (*p* = 0.576)			6.021 (*p* = 0.111)			5.681 (*p* = 0.128)
18–30	30 (55.6)	24 (44.4)		38 (70.4)	16 (29.6)		17 (31.5)	37 (68.5)	
31–40	25 (45.5)	30 (54.5)		26 (47.3)	29 (52.7)		11 (20.0)	44 (80.0)	
41–50	21 (56.8)	16 (43.2)		22 (59.5)	15 (40.5)		4 (10.8)	33 (89.2)	
51+	6 (42.9)	8 (57.1)		8 (57.1)	6 (42.9)		3 (21.4)	11 (78.6)	
Education			8.897 (*p* = 0.012)			26.739 (*p* < 0.001)			7.535 (*p* = 0.023)
High school or lower	30 (57.7)	22 (42.3)		40 (76.9)	12 (23.1)		18 (34.6)	34 (65.4)	
Associate or bachelor’s degree	50 (54.3)	42 (45.7)		54 (58.7)	38 (41.3)		14 (15.2)	78 (84.8)	
Above bachelor’s degree	3 (17.6)	14 (82.4)		1 (5.9)	16 (94.1)		3 (17.6)	14 (82.4)	
Occupation			9.581 (*p* = 0.008)			17.073 (*p* < 0.001)			1.463 (*p* = 0.481)
Employer	8 (72.7)	3 (27.3)		4 (36.4)	7 (63.6)		2 (18.2)	9 (81.8)	
Professional	10 (29.4)	24 (70.6)		11 (32.4)	23 (67.6)		5 (14.7)	29 (85.3)	
Others	65 (56.0)	51 (44.0)		80 (69.0)	36 (31.0)		28 (24.1)	88 (75.9)	
Income			1.046 (*p* = 0.306)			0.499 (*p* = 0.480)			5.965 (*p* = 0.015)
Adequate	38 (47.5)	42 (52.5)		45 (56.3)	35 (43.8)		11 (13.8)	69 (86.3)	
Not adequate	45 (55.6)	36 (44.4)		50 (61.7)	31 (38.3)		24 (29.6)	57 (70.4)	

ATB = Antibiotics.

**Table 4 antibiotics-12-01312-t004:** Odds ratio (OR) of having a better knowledge of antibiotics, better antibiotic resistance knowledge, and antibiotic usage.

Variables	ATB Knowledge	ATB Resistance	ATB Use
	OR (95% CI)	OR (95% CI)	OR (95% CI)
Gender (Ref: Female)			
Male	0.58 (0.29–1.13)	0.57 (0.27–1.19)	0.72 (0.31–1.67)
Age (Ref: 18–30)			
31–40	1.03 (0.44–2.43)	1.48 (0.59–3.67)	1.16 (0.42–3.19)
41–50	0.75 (0.30–1.88)	1.23 (0.46–3.28)	3.25 (0.93–11.35)
51+	1.74 (0.49–6.11)	1.98 (0.53–7.43)	1.46 (0.33–6.44)
Education (Ref: High school or lower)			
Bachelor’s degree and above	1.13 (0.54–2.40)	2.53 (1.09–5.84) *	3.01 (1.26–7.20) *
Occupation (Ref: Employer)			
Professional	6.06 (1.23–29.82) *	0.88 (0.19–4.00)	1.27 (0.17–9.36)
Others	2.12 (0.49–9.21)	0.27 (0.07–1.09)	1.05 (0.17–6.35)
Income (Ref: Not adequate)			
Adequate	1.39 (0.73–2.74)	1.08 (0.53–2.20)	2.66 (1.13–6.29) *

* *p* < 0.05; ATB = Antibiotics; OR = Odd ratio; Ref = Reference group.

## Data Availability

The data presented in this study are available on request. The data are not publicly available due to patient privacy. Informed consent was obtained from all subjects involved in the study. Written informed consent has been obtained from the patient(s) to publish this paper.

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
