# Peer review of "Antibiotic Knowledge, Antibiotic Resistance Knowledge, and Antibiotic Use: A Cross-Sectional Study among Community Members of Bangkok in Thailand"

_antibiotics, 2023, doi:10.3390/antibiotics12081312_

Round 1
Reviewer 1 Report
My concern is, how the chosen 161 persons represent all Bangkok residents.

Sometimes it is difficult to understand what the authors want to tell... I show a few examples in the attached document.
Author Response
Reviewer I
Title: Antibiotic knowledge, antibiotic resistance knowledge, and antibiotic use: A cross-sectional study among community member of Bangkok in Thailand
Abstract
Keywords:
Introduction
Row 52 In Thailand, at least 88,000 infected people are attributed ….better exposed to….antibiotic resistance, and approximately 38,000 people die.
Thank you so much for your comment. We have changed “attributed” to “exposed” in the sentence.
Row 58 The data showed that Thai residents spent 0.9% of what? Their income? on purchasing antibiotics in 2013, and it increased to 1.6% in 2014.
Thank you for your comment. We have added “of their expenses”.
Row 60 aimed at increasing…to increase.. public awareness of antibiotic use and tackling antibiotic resistance between 2016-2021
Thank you so much. We have rewritten the sentence as follow: In 2016, the Thai government endorsed its first National Strategic Plan 2017-2021, which was aligned with a Global Action Plan developed by WHO in 2015 to promote public awareness of antibiotic use and tackle antibiotic resistance.
We changed “from” to “between”.
Row 66: populous …populated… city of Thailand
Thank you for your comment. We have changed “populous” to “populated”.
Results
Row 72 had a higher proportion of … were…individuals who lived
Thank you so much for your suggestion. We deleted “had a higher proportion of” and changed to “were”.
Row 124 Bangkok residents have used antibiotics as follows: 37.9% in the last six months, 28% in the last 30 days, and 12.4% in the last year.
Thank you for your comment. We added “as follows” in the sentence.
Discussion
Thank you so much for your comment. As we wrote the discussion section, we attempted to provide clear explanations to the reader/reviewer.
Methods
Row 220 entitled "how s….
Thank you so much. We changed “titled” to “entitled”.
Conclusions are ok.
Thank you so much.
Tables are ok.
Thank you so much.
References
Ref.7, 10, 19, 22, 27, 35- pages are missing
Thank you so much. We checked the reference and corrected any missing pages.

Reviewer 2 Report
This study seems to have a rather small sample size. Please provide rationale for such small sample size, i.e. sample size calculation. Also, repeated statistical analysis increases odds of having a false positive, therefore the p value should be somewhat adjusted. Bot of these issues should be addressed in discussion of the Manuscript as great limitations of the study.
Table 1 should state Age (years)
Figure 1 and 2 resolution should be improved
Author Response
Reviewer II
This study seems to have a rather small sample size. Please provide rationale for such small sample size, i.e. sample size calculation. Also, repeated statistical analysis increases odds of having a false positive, therefore the p value should be somewhat adjusted. Bot of these issues should be addressed in discussion of the Manuscript as great limitations of the study.
Thank you so much for your comment. We have added 4.2 Sample Size on page 10. Our additions are as follows:
4.2 Sample Size
An estimation of the sample size for logistic regression analysis was based on the formula n = 100 + xi, where x is an integer and i is the number of independent variables [39]. An event per variable (EPV) of 10 was selected, and five sociodemographic variables were tested in the logistic model, so a minimum sample size of 150 was required. For this study, 161 participants would be an adequate sample size. The formula to calculate sam-ple size is shown below.
n = 100 + xi
where n = sample size
x = integer (x = 10)
i = independent variable (i = 5)
Table 1 should state Age (years)
Thank you so much. We have added “years”.
Figure 1 and 2 resolution should be improved
We apologize for the blurry vision in figures 1 and 2. Both figures have been revised and added to the manuscript.

Reviewer 3 Report
Thank you for this paper which I read with interest. I have a number of suggestions to make to enhance the content of the paper. These include more general remarks as well as specific remarks.
A) General - Where possible I would contrast Thailand to LMICs only and not high income countries as the circumstances are very different. This is because the purchasing of antibiotics without a prescription is appreciably higher in LMICs with often high patient co-payments and no UHC. In addition, consumption of antibiotics in ambulatory care likely to be nearer 90 - 95% of total human consumption (Duffy E et al. Antibacterials dispensed in the community comprise 85%-95% of total human antibacterial consumption. J Clin Pharm Ther. 2018;43:59-64)
B) Specific
i) Recent references regarding the global extent of AMR is Global burden of bacterial antimicrobial resistance in 2019: a systematic analysis. Lancet. 2022;399:629-55. The two references quoted (5/6) supposedly from the WHO quoting 2019 figures were were published before 2019! This needs to change. The costs of AMR can be considerable and rising Hofer U. The cost of antimicrobial resistance. Nat Rev Microbiol. 2019;17:3; Dadgostar P. Antimicrobial Resistance: Implications and Costs. Infect Drug Resist. 2019;12:3903-10. These concerns led to the WHO to launch the Global Action Plan to reduce AMR (WHO. GLOBAL ACTION PLAN ON ANTIMICROBIAL RESISTANCE - https://apps.who.int/iris/bitstream/handle/10665/193736/9789241509763_eng.pdf?sequence=1) - which resulted in national action plans. This is because if AMR not contained will become the next pandemic - Gautam A. Antimicrobial Resistance: The Next Probable Pandemic. JNMA J Nepal Med Assoc. 2022;60:225-8. Hence studies such as these! Good to know some of the activities/ challenges in Thailand and potential ways forward that this study can help with - Sumpradit N et al. Thailand's national strategic plan on antimicrobial resistance: progress and challenges. Bull World Health Organ. 2021;99(9):661-73
ii) We know nothing about the situation regarding the availability of antibiotics in Thailand in ambulatory care and any measures to reduce inappropriate prescribing/ dispensing as a prelude to why this study. Some of this information is contained within the Discussion - lines 200 - 204.
iii) Lines 45/ 46 - I would quote the actual references within 8/9 - and not just the papers that contain these references
iv) Lines 156 to 158 should be in the Introduction and not in the Discussion. The Discussion should just focus on the findings - and any similarities/ differences with other LMICs and why - and the implications building on the challenges, etc., with the NAP (Sumpradit et al 2021). The implications consolidated in the updated Conclusion
Overall reasonable
Author Response
Reviewer III
Thank you for this paper which I read with interest. I have a number of suggestions to make to enhance the content of the paper. These include more general remarks as well as specific remarks.
Thank you for your kind comments.
- A) General - Where possible I would contrast Thailand to LMICs only and not high income countries as the circumstances are very different. This is because the purchasing of antibiotics without a prescription is appreciably higher in LMICs with often high patient co-payments and no UHC. In addition, consumption of antibiotics in ambulatory care likely to be nearer 90 - 95% of total human consumption (Duffy E et al. Antibacterials dispensed in the community comprise 85%-95% of total human antibacterial consumption. J Clin Pharm Ther. 2018;43:59-64).
Thank you for your suggestion. We have added information to compare Thailand with other LMICs in the first paragraph on page 2 as follows: “Antibiotic resistance is a major problem in low- and middle-income countries (LMICs) due to the widespread use of antibiotics”…..More than…. “In Vietnam and Bangladesh, for instance, the percentage of antibiotics used without a prescription range from 38.0%-55.2% and 45.7%, respectively”.
- B) Specific
- i) Recent references regarding the global extent of AMR is Global burden of bacterial antimicrobial resistance in 2019: a systematic analysis. Lancet. 2022;399:629-55. The two references quoted (5/6) supposedly from the WHO quoting 2019 figures were were published before 2019! This needs to change.
Thank you so much for your comment. We have made the proper modification for better understanding on page one lines 43-46 as follows: “In 2019, a systematic analysis of the global burden of bacterial antimicrobial resistance (AMR) among 204 countries and territories revealed that 4.95 million deaths were related to bacterial AMR, of which 1.27 million were caused by bacterial resistance”.
The costs of AMR can be considerable and rising Hofer U. The cost of antimicrobial resistance. Nat Rev Microbiol. 2019;17:3; Dadgostar P. Antimicrobial Resistance: Implications and Costs. Infect Drug Resist. 2019;12:3903-10. These concerns led to the WHO to launch the Global Action Plan to reduce AMR (WHO. GLOBAL ACTION PLAN ON ANTIMICROBIAL RESISTANCE - https://apps.who.int/iris/bitstream/handle/10665/193736/9789241509763_eng.pdf?sequence=1) - which resulted in national action plans. This is because if AMR not contained will become the next pandemic - Gautam A. Antimicrobial Resistance: The Next Probable Pandemic. JNMA J Nepal Med Assoc. 2022;60:225-8. Hence studies such as these! Good to know some of the activities/ challenges in Thailand and potential ways forward that this study can help with - Sumpradit N et al. Thailand's national strategic plan on antimicrobial resistance: progress and challenges. Bull World Health Organ. 2021;99(9):661-73
Thank you so much for your comment. We have made the changes as follows:
- On page two lines 47-51-healthcare cost in different regions: “Antibiotic resistance has an impact not only on human health and lives but also on over-all healthcare costs. There is an estimate that antibiotic resistance will result in healthcare expenditures of 20 billion dollars per year in direct healthcare costs in the United States and up to 3.5 billion dollars annually in Europe, North America, and Australia”.
- On page two lines 54-56-healthcare cost in Thailand: “The direct costs associated with AMR in 2010 were about 70-170 million dollars, while indirect costs for morbidity and premature death totaled 1,100 million dollars”.
- On page two lines 70-79-activities/challenges in Thailand: In 2016, the Thai government endorsed its first National Strategic Plan 2017-2021, which was aligned with a Global Action Plan developed by WHO in 2015 to promote public awareness of antibiotic use and tackle antibiotic resistance. Over the three-year period from 2017-2019, knowledge of AMR and antibiotic use increased by 0.6% from 23.7% to 24.3%, while human antibiotic consumption increased by 20.9% from 68.4% to 83.0% defined daily dose per 1,000 inhabitants per day, in comparison to the goal of reducing antibiotic consumption by 20% and improving antibiotic knowledge and antibiotic resistance by 20% [18, 19]. Indeed, inadequate knowledge of antibiotics is associated with irrational use of antibiotics within the community, resulting in antibiotic resistance”.
- ii) We know nothing about the situation regarding the availability of antibiotics in Thailand in ambulatory care and any measures to reduce inappropriate prescribing/ dispensing as a prelude to why this study. Some of this information is contained within the Discussion - lines 200 - 204.
Thank you for your suggestion. We have improved the paragraph for better understanding about availability of antibiotics in Thailand on page two lines 68-70 as follows: “Likewise, a licensed pharmacist can legally dispense antibiotics in Thailand, where residents can purchase most antibiotics from pharmacies without a prescription, potentially resulting in overuse or inappropriate use of antibiotics”.
iii) Lines 45/ 46 - I would quote the actual references within 8/9 - and not just the papers that contain these references
Thank you for your comment. We have changed to the actual references from 8/9 [(8) Al Rasheed, A., et al., Prevalence and Predictors of Self-Medication with Antibiotics in Al Wazarat Health Center, Riyadh City, KSA. Biomed Res Int, 2016. 2016: p. 3916874. and (9) Alhomoud, F., et al., Self-medication and self-prescription with antibiotics in the Middle East-do they really happen? A systematic review of the prevalence, possible reasons, and outcomes. Int J Infect Dis, 2017. 57: p. 3-12) to 14/15 [(14) Cars, O. and P. Nordberg, Antibiotic resistance–The faceless threat. International Journal of Risk & Safety in Medicine, 2005. 17(3-4): p. 103-110. and (15) Morgan, D.J., et al., Non-prescription antimicrobial use worldwide: A systematic review. Lancet Infect Dis, 2011. 11(9): p. 692-701). Also, line 45/46 has been changed to line 64/65 to make the sentence more fluent.
- iv) Lines 156 to 158 should be in the Introduction and not in the Discussion. The Discussion should just focus on the findings - and any similarities/ differences with other LMICs and why - and the implications building on the challenges, etc., with the NAP (Sumpradit et al 2021). The implications consolidated in the updated Conclusion.
Thank you so much for your comment. We deleted the first paragraph and added a new short paragraph on page eight lines 184-185 as follows: “This study provides insights into antibiotic knowledge, antibiotic resistance knowledge, and antibiotic usage within particular residents in Bangkok” to lead the readers to the findings.
We have also removed the statement “In contrast, only 15% of antibiotic usage is found in high income countries such as Ger-many and Sweden” so that the reader can focus on only lower-middle income countries.
We have also added the statement “Clearly, antibiotics overuse is very challenging in Thailand, where there is a need for political commitment, regular structure, financial resources, data availability on antibiotic use, overuse, and misuse, and national drug policy to sustainably tackle antibiotic resistance” to refer to antibiotics overuse and what Thailand needs.

Round 2
Reviewer 3 Report
Thank you for the update. I have no further suggestions to make and believe the revised paper should now be accepted for publication.